# Development and Validation of a Novel Four Gene-Pairs Signature for Predicting Prognosis in DLBCL Patients

**DOI:** 10.3390/ijms252312807

**Published:** 2024-11-28

**Authors:** Atsushi Tanabe, Jerry Ndzinu, Hiroeki Sahara

**Affiliations:** 1Laboratory of Highly-Advanced Veterinary Medical Technology, Veterinary Teaching Hospital, Azabu University, 1-17-71 Fuchinobe Chuo-ku, Sagamihara 252-5201, Kanagawa, Japan; 2Laboratory of Biology, Azabu University School of Veterinary Medicine, 1-17-71 Fuchinobe Chuo-ku, Sagamihara 252-5201, Kanagawa, Japan; jerry@azabu-u.ac.jp (J.N.); sahara@azabu-u.ac.jp (H.S.); 3Department of Research and Development (R&D), Malignant Tumor Treatment Technologies, Inc., 130-42 Nagasone, Kita-ku, Sakai 591-8025, Osaka, Japan

**Keywords:** diffuse large B-cell lymphoma, public database, gene signature, prognosis, drug sensitivity

## Abstract

Diffuse large B-cell lymphoma (DLBCL) is the most common subtype of non-Hodgkin’s lymphoma. Because individual clinical outcomes of DLBCL in response to standard therapy differ widely, new treatment strategies are being investigated to improve therapeutic efficacy. In this study, we identified a novel signature for stratification of DLBCL useful for prognosis prediction and treatment selection. First, 408 prognostic gene sets were selected from approximately 2500 DLBCL samples in public databases, from which four gene-pair signatures consisting of seven prognostic genes were identified by Cox regression analysis. Then, the risk score was calculated based on these gene-pairs and we validated the risk score as a prognostic predictor for DLBCL patient outcomes. This risk score demonstrated independent predictive performance even when combined with other clinical parameters and molecular subtypes. Evaluating external DLBCL cohorts, we demonstrated that the risk-scoring model based the four gene-pair signatures leads to stable predictive performance, compared with nine existing predictive models. Finally, high-risk DLBCL showed high resistance to DNA damage caused by anticancer drugs, suggesting that this characteristic is responsible for the unfavorable prognosis of high-risk DLBCL patients. These results provide a novel index for classifying the biological characteristics of DLBCL and clearly indicate the importance of genetic analyses in the treatment of DLBCL.

## 1. Introduction

Non-Hodgkin’s lymphoma (NHL) refers to a group of blood cancers that arise from lymphoid tissue, primarily lymph nodes. NHL is a highly heterogeneous disease composed of various subtypes exhibiting diverse clinical and molecular features. Diffuse large B-cell lymphoma (DLBCL) is the most common subtype of NHL and has its own highly heterogeneous biology. Standard primary treatment for DLBCL is R-CHOP, a combination of rituximab, cyclophosphamide, doxorubicin, vincristine, and prednisone. Although addition of the anti-CD20 antibody rituximab to standard therapy has greatly improved DLBCL outcomes, approximately 40% of patients may relapse or become refractory to therapy [1,2]. Various salvage therapies for relapsed/refractory DLBCL are currently being developed and investigated, but few patients achieve long-term remission [2,3].

The International Prognostic Index (IPI), which classifies DLBCL patients into different risk groups based on five factors (age, serum LDH, performance status, stage, and number of extranodal lesions), is the most prevalent prognostic predictor for patients with DLBCL, but the IPI alone does not describe the biological characteristics of DLBCL [4,5]. Since the late 1990s, various classification models using gene-expression profiling as a predictor have been developed in an effort to clarify the biological heterogeneity of DLBCL. The most representative and clinically utilized classification model is cell of origin (COO) subtyping, which classifies DLBCL into two groups: germinal-center B-cell (GCB) type and activated B-cell (ABC) type [6]. In general, the prognosis of ABC-DLBCL is worse than that of GCB-DLBCL. Classification models that focus on identifying patterns of chromosomal translocations and specific gene mutations have also been developed in response to recent advancements in next-generation sequencing techniques [7,8]. In addition to determining the genetic characteristics of lymphoma cells, classification models have also been developed to characterize components of the lymphoma microenvironment, such as stromal cells and immune cells [9,10]. However, various improvements must be instituted in order to render these prognostic predictors useful in the patient care context, such as improving their accuracy and reproducibility, expanding their versatility, and reducing the clinical testing cost.

In the present study, we conducted a comprehensive analysis of gene-expression profiles in public databases to identify novel gene signatures for classifying the biological characteristics and prognosis of DLBCL, using a univariable Cox proportional hazard model to extract genes relevant to prognosis of DLBCL. A risk-scoring model we developed showed a superior predictive stability compared to other existing risk-scoring models. In addition, this risk-scoring model was found to be clearly associated with chemotherapy resistance of DLBCL.

## 2. Results

### 2.1. The Development of Four Gene-Pair Signature Model

To identify novel prognostic gene signatures for DLBCL, we examined eight public mRNA-expression datasets obtained from two public microarray platforms (five datasets from Affymetrix, three datasets from ILMN) (Figure 1). To correct for batch effects between datasets, datasets from the same platform were merged into DLBCL training dataset 1 and 2 using pyComBat (Appendix A). After extracting data for patients treated with CHOP or R-CHOP from the integrated data, univariable Cox regression analysis was performed to identify prognostic genes associated with overall survival of DLBCL patients. Finally, 408 prognostic genes were identified based on overlap between the two training datasets (Figure 2A,B and Appendix A). To examine the relationships between the 408 newly identified prognostic genes and existing prognostic genes for DLBCL, a literature review was conducted. When 408 prognostic genes were matched to 4575 DLBCL-related genes in signatureDB (http://lymphochip.nih.gov/signaturedb/, accessed on 13 September 2024) [11], 242 genes overlapped (Appendix A). In addition, 48 genes overlapped with the 777 genes in the BLYM-777 gene panel, which is based on more than 45 DLBCL studies [12].

Next, these 408 prognostic genes were combined for a total of 41,616 gene-pairs. To screen for gene-pairs associated with the prognosis of DLBCL patients, we performed univariate Cox regression analysis again. A total of four gene-pairs were common to training datasets (Figure 2C–K, and Appendix A). Detailed information on the seven prognostic genes in the four gene-pairs is provided in Table 1. Finally, a risk score was calculated by directly summing the binary scores of the four gene-pairs.

### 2.2. Assessing the Performance of the Risk Score as a Prognostic Biomarker

To investigate predictive performance of the risk score, DLBCL patients in training datasets were divided into five groups based on the risk score. The KM survival curves showed that the OS was clearly reduced in the high-risk score group (Figure 3A,B). To confirm predictive performance of the risk score, we analyzed additional DLBCL datasets. Due to the lack of information on prognostic genes, the risk scores in GSE4475, GSE32918, and GSE68895 datasets were calculated using three, two, and two gene-pairs, respectively (Appendix A). The KM survival curves showed that DLBCL patients in high-risk group had an unfavorable prognosis than those in the low-risk groups (Figure 3C,D and Appendix A). In addition, many patients with primary central nervous system lymphoma with unfavorable prognosis showed high-risk scores (Appendix A). These results suggested that the risk score based on the four gene-pairs is a useful biomarker for classifying prognostic differences among DLBCL patients.

### 2.3. Predictive Performance of the Risk Score in Conditions Stratified by Clinical Characteristics

To investigate the relationship between the risk score and clinical characteristics of DLBCL, we examined predictive performances of the risk score in DLBCL patients with different clinical characteristics. In this analysis, all of DLBCL patients in GSE117556 and GSE181063 datasets were integrated into one dataset. First, we compared the predictive performance of the risk score in DLBCL patients in Low, Intermediate, and High IPI-scores (Figure 4A–D). Compared to the Low-IPI score group, patients in the High-IPI score group tended to have a higher percentage of high-risk scores. In all three groups stratified by IPI-scores, DLBCL patient with high-risk score exhibited significantly worse outcomes (Figure 4B–D). Retrospective analysis has divided DLBCL patients in GSE117556 and GSE181063 into four molecular subtypes: GCB, ABC, and unclassified (UNC), which are common COO subtypes, and molecular high-grade (MHG), which has a worse prognosis (Figure 4E). Notably, the risk score was also independent prognostic predictor in different molecular subtypes (Figure 4F–I). DLBCL harboring rearrangements in Myc and Bcl2 or Bcl6, referred to as double-hit lymphoma, exhibits a distinctly worse prognosis (Figure 4J). DLBCL patients with high-risk score had a worse prognosis in both the no Myc rearrangement group and the double-hit group. (Figure 4K,L). These results indicated that the risk score remained a useful prognostic factor even after stratification by various clinical characteristics.

### 2.4. Comparison Predictive Performance of the Four Gene-Pair Signatures Against Other Signatures

To verify robustness of the four gene-pair signatures, we compared predictive performance against nine existing gene signatures [13,14,15,16,17,18,19,20,21]. The gene symbols and regression coefficients for the nine signatures are listed in Appendix A. Compared to the distribution of risk scores for the four gene-pair signatures, the distribution of risk scores for the other signatures was highly variable across DLBCL datasets (Appendix A). To evaluate the predictive accuracy of the signatures, we calculated their area under the curve (AUC) scores in six external validation datasets and one internal test datasets using time-dependent ROC curve analyses (Figure 5, Appendix A). The four gene-pair signatures showed the best AUC score in GSE98588 dataset (Figure 5, Appendix A). When averaging AUC scores at 1 to 3 years, the four gene-pair signatures showed consistently accurate predictive performances (Figure 5, Appendix A). Some of the other signatures showed relatively high prognostic accuracy in certain datasets, especially those in which they were trained, but many of them showed reduced accuracy in other datasets. For instance, the Xiong’s signature showed a high predictive accuracy in GSE11318, where it was trained, but it showed worse accuracy in other datasets (Figure 5, Appendix A). When the accuracy of the AUC score was evaluated in rank order format, the four gene-pair signature and Ren’s signature tied for first place (Table 2). These results showed that the risk-scoring models based on the four gene-pair signature provides a robust prognostic predictor for DLBCL patients.

### 2.5. Differential Signaling Pathways in Low- and High-Risk DLBCL

To explore differences in signaling pathways between low- and high-risk DLBCL, GSVA was performed. We assessed differences in the activity of Kyoto Encyclopedia of Genes and Genomes (KEGG) pathways in the low- and high-risk groups. The GSVA scores of 85 pathways was significantly different between low- and high-risk DLBCL samples in training datasets (Figure 6, Appendix A). Pathways involved in cell-cell interactions with stromal and immune cells and drug metabolism pathways were relatively activated in low-risk DLBCL. By contrast, cell cycle, energy synthesis, and DNA repair pathways were activated in high-risk DLBCL. The GSVA results suggested that differences in these signaling pathways might alter resistance to chemotherapy and had a critical impact on the prognosis of DLBCL patients.

### 2.6. High-Risk DLBCL Cell Lines Are Resistant to Chemotherapy

To examine the drug sensitivity of DLBCL associated with risk score, IC50 values of anti-cancer drugs and gene-expression data for 17 DLBCL cell lines were obtained from the Cancer Cell Line Encyclopedia and subjected to risk scoring analysis (Figure 7A). Comparative evaluation of drug sensitivity of low- and high-risk DLBCL cell lines revealed that high-risk DLBCL cell lines were significantly more resistant to 12 drugs and tended to be resistant to 14 drugs (Table 3). These drugs included doxorubicin, cisplatin, etoposide, and gemcitabine, all of which play a central role in chemotherapy of DLBCL (Figure 7B–D, Table 3). By contrast, there was no clear difference in susceptibility between the low- and high-risk groups for cytarabine and methotrexate (Figure 7E,F). Unfortunately, the comparative analysis did not find any drugs that were specifically effective against high-risk DLBCL cell lines (Table 3). DLBCL cell lines were also divided into low- and high-risk groups based on the Ren’s signature, which has high predictive accuracy, and then comparative evaluation of drug sensitivity was performed. However, when the cell lines were grouped by the Ren’s signature, the low-risk DLBCL cell lines were more resistant to various drugs, including cisplatin, gemcitabine, and cytarabine (Appendix A). These results suggested that the risk score based on the four gene-pair signature is strongly correlated with the chemotherapy resistance of DLBCL.

## 3. Discussion

Since the late 1990s, advances in genetic analysis technologies have led to an increase in research on the genetic characteristics of various carcinomas, including DLBCL [6,7,8]. More recent studies have used a variety of methods to take advantage of cancer transcriptome data available in public databases [9,10]. In this study, we comprehensively analyzed DLBCL transcriptome data to extract new gene signatures and classify the prognostic characteristics of DLBCL.

Among the DLBCL prognostic genes we identified, 204 were found to be associated with a favorable prognosis. By contrast, 204 genes associated with an unfavorable prognosis were also identified, including many genes involved in lymphoma growth and survival, such as Myc and Bcl2 [7,8]. From the vast combination of those prognostic genes, a four gene-pairs model consisting of seven signature genes was developed. To investigate prognostic values and biological roles of seven signature genes in DLBCL, we searched signatureDB using those signature genes as search terms. NEK6 gene has been reported to be a marker gene for GCB-type DLBCL [22]. RARRES2 gene was reported by Lenz et al. as a Stromal-1 signature gene associated with a favorable prognosis [23]. No officially published reports on the CHRNA1 gene were found in signatureDB. NEK6, URI1, NIPA2, ZNF22, and WDR12 were all included in the EcoTyper signature genes [10]. NEK6 was a signature gene of B-cell state 1 (S1) associated with a favorable prognosis, while URI1, ZNF22, NIPA2, and WDR12 were all signature genes of B-cell state 4/5 (S4/S5) associated with an unfavorable prognosis [10]. Thus, it is very likely that each of these signature genes is an important marker associated with prognosis of DLBCL, but further studies are needed to reveal the biological roles of these genes in DLBCL development and survival.

We developed a risk-scoring model based on the four gene-pairs with reference to a predictive model for acute myeloid leukemia published by Kong et al. in 2022 [24]. In the predictive model of Kong et al., the risk score was calculated after multiplying the binary scores of the gene-pairs by the regression coefficient, whereas in our risk-scoring model, the risk score was calculated as it is in the binary scores. Since the final risk score has five levels from 0 to 4, it is easier to determine the low-risk and high-risk threshold than in other risk-scoring models. The risk-scoring model based on the four gene-pairs showed stable accuracy as a prognostic predictor for DLBCL patients in training and validation datasets. The risk-scoring model was developed based on gene-expression data from DLBCL patients receiving standard R-CHOP/CHOP therapy, but this model also showed high predictive accuracy in a group of patients not receiving standard chemotherapy. Furthermore, it was shown to be an independent prognostic predictor even when combined with clinicopathologic predictors such as IPI-score, COO subtype, and myc-rearrangement.

To confirm the robustness of our risk-scoring model, we compared AUC scores in external validation datasets with other existing predictive models. The results showed that our risk-scoring model outperformed almost all other predictive models in terms of predictive stability. Among the other nine models, Ren’s signature model was found to be an excellent predictive model, showing equal or more stable predictive accuracy than our risk-scoring model in all six datasets except TCGA-DLBC. However, the fact that Ren’s signature consists of 24 signature genes is a point that makes it inferior to other predictive models when considering the operational cost of measuring gene expression.

The GSVA results showed that cell-cell interactions with stromal and immune cells were activated in low-risk DLBCL. Kotlov and colleagues described four DLBCL microenvironments based on analyses of transcriptome data from more than 4000 DLBCL cases [9]. Of the four microenvironments, “Depleted-DLBCL”, characterized almost exclusively by B-lymphoma cells and very few other normal stromal and immune cells, had the most unfavorable prognosis. The results of the analysis by Kotlov et al. [9] strongly suggest that interactions between DLBCL and normal stromal and immune cells have a major effect on DLBCL survival and proliferation. The GSVA results also showed that several drug metabolic pathways were activated in low-risk DLBCL. Cyclophosphamide in R-CHOP therapy is a prodrug, which is metabolized in the liver to the active form [25,26]. The activation of drug metabolic pathways in low-risk DLBCL suggested that the metabolism of cyclophosphamide within the tumor might affect sensitivity to R-CHOP therapy. In contrast, the GSVA results showed that DNA repair pathways were activated in high-risk DLBCL. The activation of these pathways suggested that high-risk DLBCL are highly resistant to DNA damage caused by cyclophosphamide and doxorubicin.

To examine the association between risk score and DLBCL resistance to chemotherapy, drug resistance in low- and high-risk DLBCL cell lines was compared. In this study, the resistance to approximately 260 drugs in the CCLE dataset were compared between the two groups, and high-risk DLBCL cell lines were resistant to 26 drugs, equivalent to one-tenth of the total drugs. Especially, since doxorubicin and cisplatin are important drugs in first-line and second-line therapy for DLBCL, resistance to them might be an important factor in the unfavorable prognosis of DLBCL patients with high-risk scores. In addition, the result that high-risk DLBCL cell lines were resistant to drugs that cause DNA damage was consistent with the GSVA results. By contrast, the risk score based on Ren’s signature did not correlate with drug resistance of DLBCL cell lines. The 24 signature genes in Ren’s signature include marker genes for T-cells and dendritic cells, such as CD3E, CD28, and LY75. Thus, Ren’s signature might be a risk-scoring model that reflects the state of the DLBCL microenvironment, suggesting that the model does not work properly in cell culture systems where lymphoma cells are present alone.

One of the major clinical implications of the development of biomarkers and predictive models is to provide useful information for patient care. For example, a reliable biomarker that can be used to guide the choice of a treatment modality such as chimeric antigen receptor T cell therapy, which is highly effective but expensive and carries the risk of serious side effects, would be of great benefit to both health care workers and patients [27]. In the treatment of DLBCL, several common predictors, such as IPI score and COO subtype, help provide reliable prognostic information, but they are not often reflected in treatment modalities. On the other hand, recently developed clustering models based on the LymphGen algorithm have been shown to provide a decision index for predicting the efficacy of specific molecular-targeted drugs [7]. Our survival analysis and drug screening test showed that high-risk DLBCL was highly resistant to various chemotherapies, including R-CHOP. However, it was suggested that the sensitivity to methotrexate was almost the same in low-risk and high-risk DLBCL. Therefore, prophylactic high-dose methotrexate therapy for DLBCL patients with high-risk scores may improve prognosis [28]. In the second-line therapy for high-risk DLBCL, the usage of new molecular-targeted drugs and immunotherapies should be considered more aggressively than general treatment regimens, including cisplatin and etoposide.

Our study has important limitations. The first limitation is that the predictive performance of the risk-scoring model was demonstrated only in retrospective studies using publicly available gene-expression datasets and not in prospective studies using an original dataset. In addition, the datasets used in this study was DLBCL patient data from approximately 10 to 20 years ago. With advances in diagnostic techniques and treatment methods, the outcome of DLBCL has gradually improved since 2003, when R-CHOP therapy became the standard of care [29]. Therefore, prospective studies are required to confirm whether the risk-scoring model can adequately predict the prognosis of new patients. The second limitation is that the predictive performance of the risk-scoring model was examined using only microarray and total RNAseq-derived gene-expression data. When envisioning clinical applications of risk-scoring models, the cost of the work required for clinical testing and the financial cost to the patients are important issues to consider. Therefore, it is necessary to consider the predictive performance of alternative methods such as real-time PCR and NanoString techology, which have relatively low operational costs, instead of microarray and RNAseq, which have high operational costs. Especially, NanoString technologies is often used in translational research for clinical applications of classification models originally derived from microarray data, such as COO subtyping [12]. In any case, further clinical studies are needed to overcome the limitations of the research at this stage.

## 4. Materials and Methods

### 4.1. Datasets

A total of 19 microarray and RNA-seq datasets [23,30,31,32,33,34,35,36,37,38,39,40,41,42,43,44,45,46,47] were obtained from the NCBI Gene Expression Omnibus (GEO) [48] and the Genomic Data Commons Data Portal [49]. IC50 values of anti-cancer drugs and RNA-seq data in Cancer Cell Line Encyclopedia (CCLE) datasets were obtained from the cBioPortal for Cancer Genomics [50,51]. Details of the datasets used in this study are summarized in Table 4.

### 4.2. Identification of Prognostic Genes Using a Univariable Cox Proportional Hazard Model

Multiple independent microarray datasets were integrated into two training datasets using “pyComBat” (https://epigenelabs.github.io/pyComBat/#using-pycombat, accessed on 16 December 2023) [52] to eliminate batch effects (Appendix A). After integration, DLBCL patients treated with regimen other than CHOP/R-CHOP were removed from the training datasets and designated as internal test data to evaluate the performance of the predictive models. Next, the Python package “lifelines” (v0.28.0) was used to perform univariate Cox regression analysis of the gene expression profiles and clinical information of the training datasets. Patients were divided into a high group (top 50%) or low group (bottom 50%) based on the median expression level for each gene. Hazard ratios (HRs) were calculated by fitting Cox proportional hazard models for the high and low groups. The threshold p value for the log-rank and Wilcoxon tests was set at <0.01, and genes with a HR < 1.0 were defined as favorable prognostic genes (FPGs), whereas genes with HR > 1.0 were defined as unfavorable prognostic genes (UPGs). Prognostic genes that overlapped in both training datasets were finally selected (Appendix A).

### 4.3. Selection of Four Gene-Pairs and Calculation of the Risk Score

Data matrices of 204 FPGs and 204 UPGs were extracted from the training datasets and normalized using QuantileTransformer (output_distribution = ‘uniform’) in Python package “scikit-learn” (v1.5.1). Next, the UPGs were paired with the FPGs into a total of 41,616 gene-pairs. These pairs were binary transformed into scores of 1, if expression of the UPG was greater or equal to that of the FPG, or 0, if less. Univariate Cox regression analysis was performed using the binary scores of the 41,616 gene-pairs and clinical information from training datasets in the Python package “lifelines”. The gene-pairs with HR values > 1.8 were extracted from training datasets (Appendix A). Then, four gene-pairs consisting of seven genes that overlapped in both training datasets were selected (Table 1).Finally, a risk score was calculated from the binary scores of the four gene-pairs. Where data for all prognostic genes were not available, the risk score was calculated using only the data of the genes registered in the dataset.

Survival analysis was performed using the Python package “lifelines”. Kaplan-Meier survival curves were generated to compare the clinical outcomes between patients stratified by risk score. Log-rank *p* values < 0.05 were considered to indicate statistical significance.

### 4.4. Time-Dependent ROC Curves and GSVA Analysis

Time-dependent receiver operating characteristic (ROC) curves and gene set variation analysis (GSVA) were performed by the R libraries “survivalROC” and “GSVA” in an Rstudio environment (R version 4.2.0; R Foundation) [53,54].

### 4.5. Data Analysis Environment and Statistical Analyses

Statistical analyses and visualization of data in the present study were primarily conducted using Python (version 3.9; Python Software Foundation) in a JupyterLab environment [55,56]. Two-tailed paired t-test was used for the comparison of two paired groups. Welch’s t-test was used for the comparison of two unpaired groups. *p* values < 0.05 were considered to be statistically significant.

## 5. Conclusions

Taken together, we developed a risk-scoring model based on four gene-pair signatures using gene-expression profiles of DLBCL from public databases. Prognosis prediction by the risk-scoring model was highly reproducible, and its predictive accuracy was superior to many other existing models. The risk-scoring model also reflects the resistance of DLBCL to chemotherapy, which may also be useful in making treatment decisions.

## Figures and Tables

**Figure 1 ijms-25-12807-f001:**
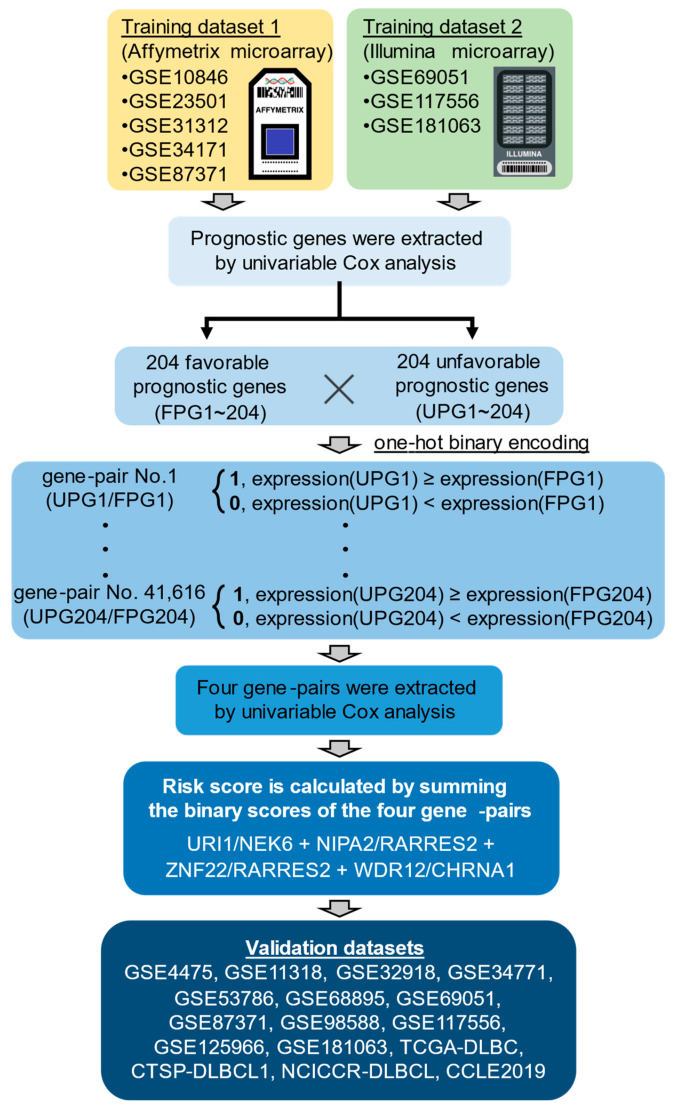
Workflow from training data integration to development of risk-scoring model. The training dataset 1 includes five Affymetrix microarray datasets and the training dataset 2 includes three Illumina microarray datasets were used for identification of prognostic genes. The risk-scoring model based on four prognostic gene-pairs was developed in the manner shown in Section 4.

**Figure 2 ijms-25-12807-f002:**
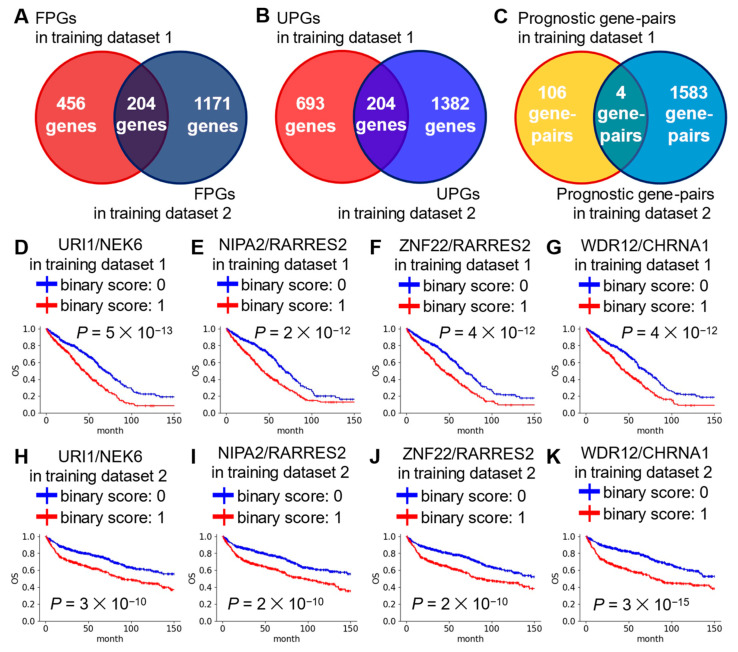
Screening of 408 prognostic genes and prognostic four gene-pairs. Venn diagrams summarizing the overlap of FPGs (**A**) or UPGs (**B**) or prognostic gene-pairs (**C**) between two training datasets. (**D**–**K**) KM survival curves showing OS in binary score groups of four gene-pairs. *p* values obtained using log-rank test.

**Figure 3 ijms-25-12807-f003:**
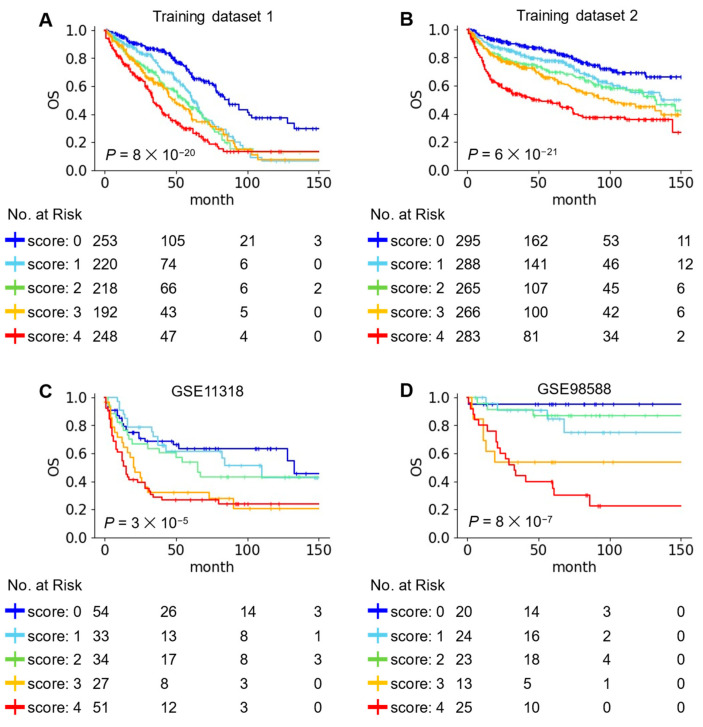
The predictive performance of the risk score in the training and validation datasets. (**A**–**D**) KM survival curves showing OS of five risk groups. *p* values obtained using log-rank test.

**Figure 4 ijms-25-12807-f004:**
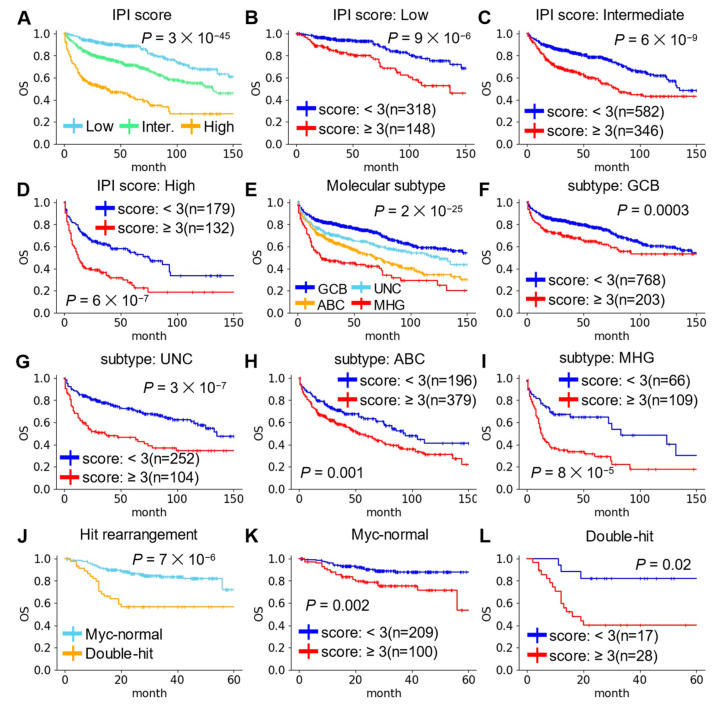
Survival analysis of the risk score in different clinical characteristics. (**A**) KM survival curves showing OS of patient with low-, intermediate-, and high-IPI-score. (**B**–**D**) KM survival curves showing OS of low- and high-risk patients in the three different IPI-score groups. (**E**) KM survival curves showing OS of patients with GCB, UNC, ABC, and MHG molecular subtypes. (**F**–**I**) KM survival curves showing OS of low- and high-risk patients in the four different molecular subtypes. (**J**) KM survival curves showing OS of patients with Myc-normal and double-hit DLBCL. (**K**,**L**) KM survival curves showing OS of low- and high-risk patients in the Myc-normal or double-hit DLBCL.

**Figure 5 ijms-25-12807-f005:**
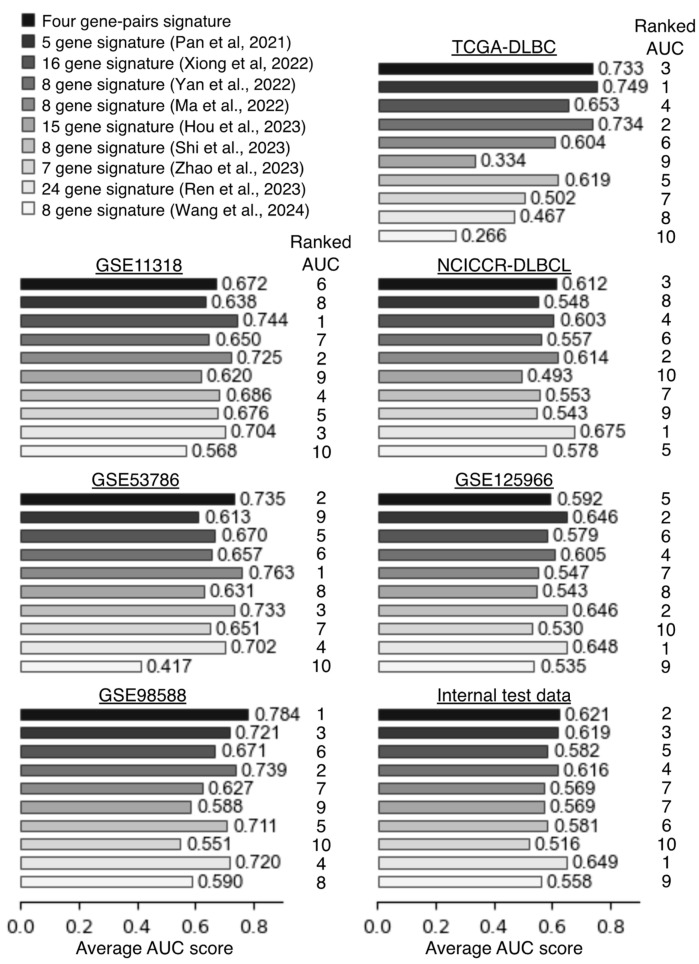
Comparison of AUC score between four gene-pair signature and nine other gene signatures. The numbers on the right side of the bar graph indicate the average AUC score at the 1-, 2-, and 3-year time points. The “Ranked AUC” is a relative ranking in predictive accuracy among 10 gene signatures [13,14,15,16,17,18,19,20,21].

**Figure 6 ijms-25-12807-f006:**
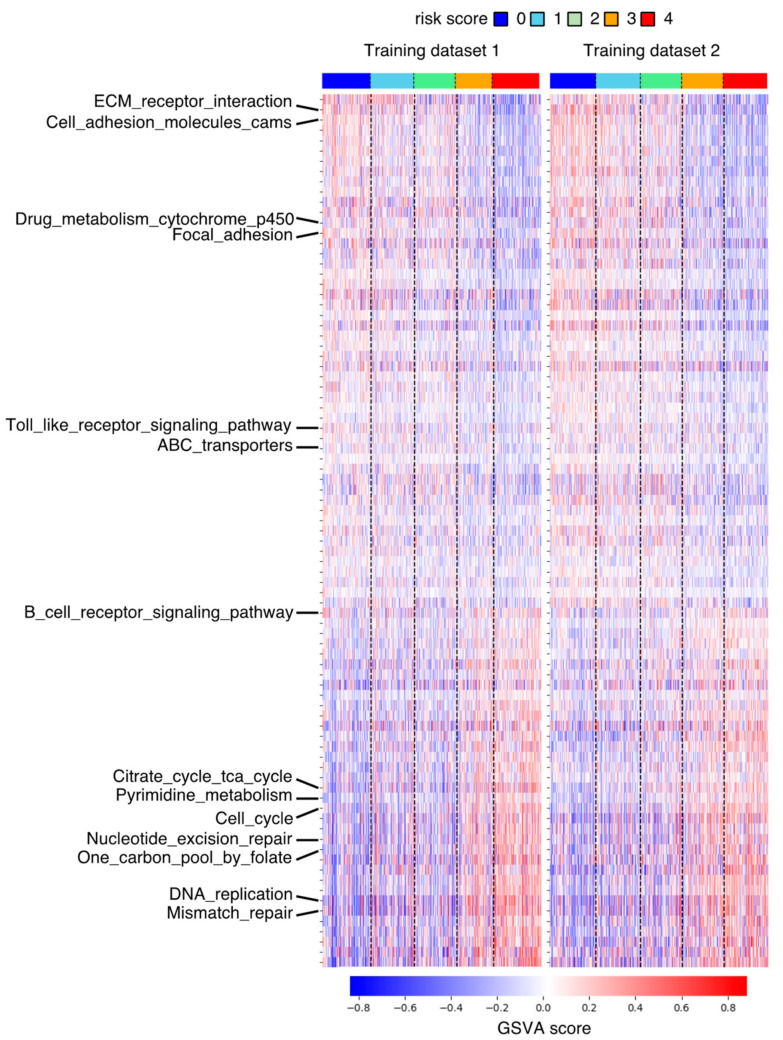
Differential signaling pathways in low- and high-risk DLBCL. DLBCL patients in two training datasets were stratified by risk score. GSVA scores of 85 KEGG pathways in each DLBCL patients in two training datasets were illustrated by heat map.

**Figure 7 ijms-25-12807-f007:**
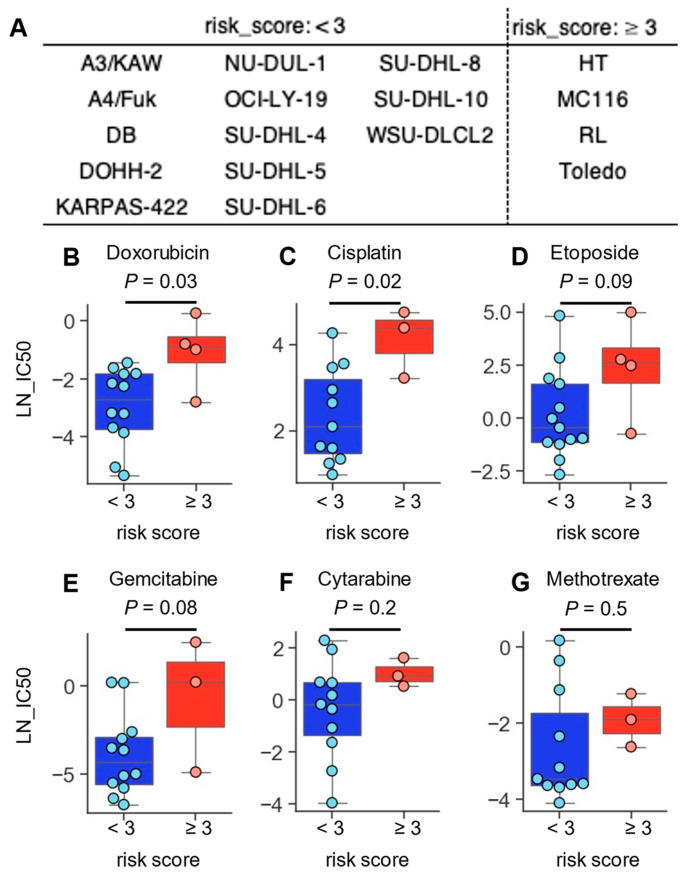
Differential drug sensitivity in low- and high-risk DLBCL cell lines. (**A**) 17 DLBCL cell lines were stratified into low- and high-risk groups by risk score (**B**–**G**) Boxplots showing log-normalized (LN) IC50 values of anti-cancer drugs in low- (blue) and high-risk (red) DLBCL cell lines. Each circle represents LN_IC50 values in an individual cell line. *p* values obtained using Weltch *t* test.

**Table 1 ijms-25-12807-t001:** Seven prognostic genes included in four gene-pairs.

Gene Symbol	Gene Name	Entrez ID	Probe ID	
			HG-U133_Plus_2	Illumina HumanHT-12 V4.0
NEK6	NIMA related kinase 6	10783	223158_s_at	ILMN_1660871
RARRES2	retinoic acid receptor responder 2	5919	209496_at	ILMN_1810844
CHRNA1	cholinergic receptor nicotinic alpha 1 subunit	1134	206633_at	ILMN_2361768
URI1(C19orf2)	URI1 prefoldin like chaperone	8725	214173_x_at	ILMN_2406892
NIPA2	NIPA magnesium transporter 2	81614	212129_at	ILMN_1720344
ZNF22	zinc finger protein 22	7570	218005_at	ILMN_1798533
WDR12	WD repeat domain 12	55759	218512_at	ILMN_1770692

**Table 2 ijms-25-12807-t002:** Comparison of ranked AUC scores between four gene-pair signature and other gene signatures.

Study	Ranked AUC Scores							
	GSE11318	GSE53786	GSE98588	TCGA-DLBC	NCICCR-DLBCL	GSE125966	Internal Test Data	Mean
Four gene-pairs	6	2	1	3	3	5	2	3.14
Pan et al., 2021 [13]	8	9	3	1	8	2	3	4.86
Xiong et al., 2022 [14]	1	5	6	4	4	6	5	4.43
Yan et al., 2022 [15]	7	6	2	2	6	4	4	4.43
Ma et al. 2022 [16]	2	1	7	6	2	7	7	4.57
Hou et al., 2023 [17]	9	8	9	9	10	8	7	8.57
Shi et al., 2023 [18]	4	3	5	5	7	2	6	4.57
Zhao et al., 2023 [19]	5	7	10	7	9	10	10	8.29
Ren et al., 2023 [20]	3	4	4	8	1	1	1	3.14
Wang et al., 2024 [21]	10	10	8	10	5	9	9	8.71

**Table 3 ijms-25-12807-t003:** Comparison of IC50 values between low- and high-risk DLBCL cell lines.

Drug	Description	LN_IC50_Mean		
		Risk Score: <3	Risk Score: ≥3	*p* Value
OSU-03012	PDK1 (PDPK1)	1.053	2.057	0.006
GSK650394	SGK2, SGK3	2.713	4.277	0.018
Cisplatin	DNA crosslinker	2.343	4.116	0.024
Doxorubicin	Anthracycline	−2.945	−1.070	0.027
Mitomycin-C	DNA crosslinker	−1.648	0.197	0.031
Bosutinib	SRC, ABL, TEC	0.887	2.463	0.035
SN-38	TOP1	−5.205	−2.933	0.035
XMD15-27	CAMK2	3.414	4.391	0.040
BX795	TBK1, PDK1 (PDPK1), IKK, AURKB, AURKC	0.973	2.161	0.042
Lestaurtinib	FLT3, JAK2, NTRK1, NTRK2, NTRK3	−1.929	−0.734	0.048
Ponatinib	ABL, PDGFRA, VEGFR2, FGFR1, SRC, TIE2,…	−1.324	0.860	0.045
ZM447439	AURKA, AURKB	1.345	2.756	0.048
Vorinostat	HDAC inhibitor Class I, IIa, IIb, IV	−0.091	0.715	0.057
BAY-61-3606	SYK	0.728	2.431	0.057
S-Trityl-L-cysteine	KIF11	0.043	1.467	0.066
HG6-64-1	BRAF	−0.219	1.562	0.069
Pelitinib	EGFR	−0.033	1.315	0.078
Gemcitabine	Pyrimidine antimetabolite	−3.887	−0.723	0.080
Tozasertib	AURKA, AURKB, AURKC, others	0.007	2.448	0.083
Tipifarnib	Farnesyl-transferase (FNTA)	0.711	2.413	0.083
ObatoclaxMesylate	BCL2, BCL-XL, BCL-W, MCL1	−2.383	−1.090	0.085
Dasatinib	ABL, SRC, Ephrins, PDGFR, KIT	−1.112	1.782	0.091
Etoposide	TOP2	0.163	2.373	0.094
Piperlongumine	Induces reactive oxygen species	1.503	2.614	0.097
CGP-60474	CDK1,CDK2,CDK5,CDK7,CDK9, PKC	−3.244	−2.015	0.097
Midostaurin	PKC, PPK, FLT1, c-FGR, others	−0.626	0.194	0.099

**Table 4 ijms-25-12807-t004:** The datasets used in this study.

Accession	Data Type	Platform	No. Samples	Ref
GSE68895	Microarray	Hu6800	77	[30]
GSE4475	Microarray	HG-U133A	127	[31]
TCGA-DLBC	RNA-seq	Illumina HiSeq 2000	47	[32]
GSE11318	Microarray	HG-U133_Plus_2	200	[33]
GSE10846	Microarray	HG-U133_Plus_2	414	[23]
GSE23501	Microarray	HG-U133_Plus_2	68	[34]
GSE31312	Microarray	HG-U133_Plus_2	470	[35]
GSE34771	Microarray	HG-U133_Plus_2	34	[36]
GSE32918	Microarray	Illumina HumanRef-8 v3.0	172	[37]
GSE34171	Microarray	HG-U133_Plus_2	68	[38]
GSE53786	Microarray	HG-U133_Plus_2	119	[39]
GSE69051	Microarray	Illumina HumanHT-12 V4.0	117	[40]
GSE87371	Microarray	HG-U133_Plus_2	223	[41]
NCICCR-DLBCL	RNA-seq	Illumina HiSeq 2500	234	[42]
GSE98588	Microarray	HG-U133_Plus_2	105	[43]
CTSP-DLBCL1	RNA-seq	Illumina HiSeq 2500	18	[44]
GSE117556	Microarray	Illumina HumanHT-12 V4.0	928	[45]
GSE125966	RNA-seq	Illumina HiSeq 2500	553	[46]
GSE181063	Microarray	Illumina HumanHT-12 V4.0	1149	[47]
CCLE 2019	RNA-seq	Illumina HiSeq 2000	17	[50]

## Data Availability

The original data presented in the study are openly available in the TCGA Research Network: https://www.cancer.gov/tcga (accessed on 22 January 2024), the GEO database: https://www.ncbi.nlm.nih.gov/geo/ (accessed on 16 December 2023), and the cBioPortal for Cancer Genomics: https://www.cbioportal.org/ (accessed on 11 November 2023).

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
