# Peer review of "Development and Validation of a Novel Four Gene-Pairs Signature for Predicting Prognosis in DLBCL Patients"

_ijms, 2024, doi:10.3390/ijms252312807_

Round 1
Reviewer 1 Report
Comments and Suggestions for Authors
The manuscript titled "Development and validation of a novel four gene-pair signature for predicting prognosis in DLBCL patients" by Atsushi Tanabe et al. aims to develop and validate an innovative gene signature based on four gene pairs to predict prognosis in patients with diffuse large B-cell lymphoma (DLBCL). Through analyses in public databases, the authors selected 408 prognostic genes and identified specific gene pairs associated with clinical outcomes.
Points to consider in the manuscript review:
- A more detailed justification for the selection of the specific gene pairs could be included, explaining how they were chosen in relation to other known prognostic genes for DLBCL.
- An additional analysis of the stability of the gene signature over time or in different patient cohorts could strengthen the validity of the results.
- Although the model uses methods like Cox regression and ROC curves for prognostic analysis, it would be relevant to discuss the inclusion of confidence intervals for risk scores, which could offer greater clarity on the variability of the results.
- I recommend presenting data distributions in boxplot format, which would facilitate visualization of the spread of risk scores among different groups.
- The study could further explore the clinical implications of the findings, addressing the potential impact of applying the model in treatment settings and highlighting possible long-term effects on patient health.
- In figure 7, experimental points could be added to the boxplot.
- It is important that the manuscript includes a discussion of the study’s limitations, such as the use of retrospective data and the absence of validation in prospective clinical trials.
It just needs minimal improvement in English
Author Response
Comments 1: A more detailed justification for the selection of the specific gene pairs could be included, explaining how they were chosen in relation to other known prognostic genes for DLBCL.
Response 1: Thank you for pointing this out. We agree with this comment. Therefore, we have added some information about the selection for the specific gene-pairs in Figure 2C and Table S2. We also have examined the relationships between the 408 prognostic genes identified in this study and known prognostic genes for DLBCL and have added some information of overlapped prognostic genes in Table S1.
Comments 2: An additional analysis of the stability of the gene signature over time or in different patient cohorts could strengthen the validity of the results.
Response 2: Thank you for pointing this out. We agree with this comment. Therefore, we have added the results of a survival analysis of other cohorts. The survival analysis for GSE68895, newly obtained from the database, has been shown in Figure S2I. And also, the survival analysis for DLBCL patients treated with R-AVCBP therapy, which were removed in the first data shaping step, has been shown in Figure S2J. (R-ACVBP: rituximab, doxorubicin, cyclophosphamide, vindesine, bleomycin, and prednisone)
Comments 3 and 4: Although the model uses methods like Cox regression and ROC curves for prognostic analysis, it would be relevant to discuss the inclusion of confidence intervals for risk scores, which could offer greater clarity on the variability of the results. I recommend presenting data distributions in boxplot format, which would facilitate visualization of the spread of risk scores among different groups.
Response 3 and 4: Thank you for pointing this out. We agree with this comment. Therefore, we have visualized the distributions of risk scores in boxplot format. Boxplots of the risk scores for four gene-pairs and other gene signatures have been shown in Figure S4.
Comments 5: The study could further explore the clinical implications of the findings, addressing the potential impact of applying the model in treatment settings and highlighting possible long-term effects on patient health.
Response 5: Thank you for pointing this out. We agree with this comment. Therefore, we have added a discussion of the clinical significance of the risk scoring model (line 286 to 302).
Comments 6: In figure 7, experimental points could be added to the boxplot.
Response 6: Thank you for pointing this out. We agree with this comment. Therefore, we have added experimental points to the boxplots in Figure 7.
Comments 7: It is important that the manuscript includes a discussion of the study’s limitations, such as the use of retrospective data and the absence of validation in prospective clinical trials.
Response 7: Thank you for pointing this out. We agree with this comment. Therefore, we have added a discussion about weak points of retrospective studies (line 306 to 310).
Reviewer 2 Report
Comments and Suggestions for Authors
The authors have analysed the publically available genetic database of 2500 Diffuse large B cell lymphoma patients and came up with 4 gene-pairs signature. These gene pairs signature predicability performances were compared with published signatures, cell culture chemotherapy results and survival data and found to have certain superiority.
The data and results were clearly presented. The argument leading to conclusion was clear. The signature deserves to be wider audience so that it can be tested prospectively in clinical trials.
A minor point:
Deliberation in the discussion section on the significance of the four gene-pairs signature, i.e. chemotherapy counselling, life expectancy prediction, etc. may further strengthen the impact of the manuscript.
Author Response
A minor point: Deliberation in the discussion section on the significance of the four gene-pairs signature, i.e. chemotherapy counselling, life expectancy prediction, etc. may further strengthen the impact of the manuscript.
Response: Thank you for pointing this out. We agree with this comment. Therefore, we have added a discussion of the clinical significance of our risk scoring model (line 286 to 302).